

# Analysis on the impact of dynamic innovation investment strategy of green supply chain enabled by blockchain

Fangfang Guo, Zhuang Wu, Yuanyuan Wang and Chenjun Liu

School of Management Engineering, Capital University of Economics and Business, Beijing, China

## ABSTRACT

**Background:** The emergence of the green supply chain represents a natural evolution from the traditional model. However, this transition has created trust concerns in operational processes. Fortunately, blockchain technology offers a promising solution to address this issue and help businesses overcome related obstacles. As artificial intelligence and blockchain continue to advance, enterprises are increasingly exploring opportunities for green innovation investments, although the optimal timing for successful product innovation can be difficult to predict.

**Methods:** The effects of successful innovation on eco-friendly supply chains are analyzed through various factors such as optimal investment strategy, level of blockchain technology, and overall system profit. Differential game theory is used to determine the most effective approach across three alliance modes: horizontal cooperative, non-cooperative, and vertical cooperative. Additionally, the impact of innovation uncertainty on member strategies and alliance selection is thoroughly examined.

**Results:** According to the results, predicting the likelihood of innovation realization can influence decision makers to prioritize current profits. Both horizontal and vertical cooperative alliance models can lead to Pareto improvements in total system profit, both before and after innovation success. However, the vertical cooperative alliance model proves to be more effective, especially at higher realization rates. Green suppliers stand to benefit from the vertical cooperative alliance model, as it can enhance their innovative investment strategy, while platform cooperation does not significantly affect their strategy. Platforms, on the other hand, can benefit from the vertical cooperative alliance model, as it can promote their innovative investment strategy and level of blockchain technology.

# INTRODUCTION

The rapid growth of the social economy has led to an increasing conflict between economic development and the availability of resources and environmental protection, which has become a significant constraint on sustainable progress. To overcome this issue, the government has introduced several green policies, including the People's Republic of China's Environmental Protection Tax Law and the People's Republic of China's Water

Corresponding author
Zhuang Wu,
wuzhuangwz11@163.com

Pollution Prevention and Control Law. The State Council has issued the Guiding Opinions on Accelerating the Establishment of a Green, Low-Carbon, and Cyclic Development Economic System, which encourages the implementation of green planning, design, investment, construction, production, living, and consumption in every aspect. The objective is to promote development based on efficient resource utilization, strict environmental protection, and effective greenhouse gas emission control, while establishing a green supply chain system to drive the transformation towards green and low-carbon energy. In response to national policies, social responsibility and environmental protection obligations, and fierce market competition, enterprises have shifted their operational strategies towards green supply chain management (*Birasnav et al., 2022*; *Roh et al., 2022*; *Le, Vo & Venkatesh, 2022*). Green supply chain management is a mode of management that focuses on environmental protection. It involves incorporations of environmental protection into all aspects of the supply chain operation and management to reduce negative impacts on the environment while optimizing supply chain operation and control to improve resource utilization efficiency and economic benefits. This process includes raw material purchasing, production and manufacturing, logistics and distribution, product sales, *etc.* The goal is to save resources and protect the environment.

In the management of green supply chains, there are various issues related to trust that need to be addressed. Information asymmetry, information opacity, counterfeiting, and selling of fake products are some of the prominent problems that are hindering the development of green supply chains. This is commonly known as the "lemon effect," which is significantly slowing down the progress of these supply chains (*Dutta et al., 2020*; *Liu et al., 2023*). With the advent of blockchain technology, the longstanding issues of trust can now be effectively addressed. Blockchain's decentralized nature, along with its transparency and non-tamperable data storage capabilities, makes it a reliable solution. Data is stored in blocks and maintained by multiple parties using cryptography to ensure secure transmission and access. This creates a robust technical system for storing data (*Obeidat et al., 2021*; *Pun, Swaminathan & Hou, 2021*). Jingdong, a Chinese e-commerce giant, has leveraged blockchain technology to solve the challenges of a green supply chain. The company has achieved the integration of "blockchain + green supply chain" through a "double-chain" fusion approach and launched the Jingdong Zhizhenchain BaaS platform to promote the development of a sustainable green supply chain. In recent times, green sustainability supply chain and green consumption trends have emerged as crucial drivers of enterprise innovation. By empowering the green supply chain with blockchain technology, inter-chain information sharing can be made transparent and traceable, thereby addressing the "trust" issue in green supply chain management. This move has caught the attention of both the business and academic communities (*Li et al., 2022*; *Feng, Lai & Zhu, 2022*; *Zhu, Sarkis & Lai, 2012*). In summary, the research on investment strategy for green supply chain innovation and blockchain technology, under the empowerment of blockchain, has significant scientific research and practical application value.

It is important to take note that businesses are currently facing tremendous pressure to survive in an unpredictable and ever-changing market environment. With the future development of the market being highly uncertain and volatile, the challenge of how to generate profits in such an environment has become a major concern for businesses. To grow and thrive in such a dynamic environment, businesses must innovate (*Pardo-del Val et al., 2014*; *Shockley & Turner, 2016*; *Watson, Senyard & Dada, 2020*; *Zheng, Ji & Su, 2020*). Constantly developing new and innovative products and having excellent products are essential for a brand to maintain a good reputation (*Han, Zhang & Liu, 2022*). Enterprises are always concerned about how to formulate a reasonable investment strategy for innovation. Investing in innovative products is one way for enterprises to achieve this goal. Research and development of new products, innovation, and technological upgrades can not only enhance the enterprise's production line process and resource utilization but can also improve the efficiency of the enterprise's services. Effective product innovation and technological upgrades can also bring new opportunities for the green marketing environment (*Del Monte & Papagni, 2003*). In July 2022, Starbucks, the coffee chain giant, introduced two classic iced coffees in the market. These products were a part of their coffee milk tea and milk tea eight treasure porridge range. The introduction of these products broke the "spice innovation" trend that had been dominating the ready-to-drink beverage market for a long time. These products gained a positive response from the market in just a few weeks, proving the "listing is fire" concept right. The introduction of innovative products can significantly impact the entire green supply chain. Companies must possess the capability to anticipate innovation to face future uncertainty. In the rapidly changing market environment, green supply chain members must adjust their strategies and operational modes to adapt to the changes in market demand by anticipating future product innovation success events. It is essential for the long-term development of green supply chain members to consider innovation investment strategies in anticipation of innovations. They should also consider the formulation of blockchain technology by distinguishing between pre-innovation and post-innovation environments.

This article explores the innovation investment strategy and cooperative alliance mode selection among green supply chain members under anticipated innovation. The goal is to enhance resource allocation across varying environments before and after green product innovation. Green suppliers use green manufacturing processes, green procurement, and select green materials for producing and innovating green products. Platforms choose eco-friendly green packaging and centralized distribution with environmentally sound transportation routes. Additionally, a platform's blockchain technology offers consumers a guarantee of the authenticity of green products. The study utilizes the stochastic stopping problem to pinpoint the timing of successful innovative products and constructs a differential game model for green supply chain members operating in cooperative alliance modes.

The article's innovative contributions include:

- Previous studies on "green supply chain + blockchain" have mostly relied on empirical analysis and static classifications, with little emphasis on theoretical scrutiny. However,

theoretical analyses can help companies predict the impact of environmental factors on their strategic operations and offer valuable management perspectives.

- Blockchain empowerment for green supply chain optimization: integrating innovation factors for better management outcomes.
- Using the random stopping problem is a more realistic way to depict uncertainty in successful product innovation timing.
- We examine how blockchain and innovation affect the selection of cooperative alliance modes in green supply chains, including horizontal and vertical alliances.

This article presents a framework for a green supply chain, including a green supplier and two internet platforms. It constructs differential game models under three different modes: horizontal cooperation, non-cooperation, and vertical cooperation. The article analyzes the dynamic strategy formulation of green supply chain members and the cooperative alliance mode choices under the influence of innovation factors. The article is structured as follows: "Related Work" presents relevant studies; "Problem Description and Model Assumptions" explains the conceptual model; "Model Analysis" examines optimal gaming models under various decision-making modes; and "Model Analysis" further investigates optimal innovation investment strategies, blockchain technology levels, and enterprise performance under different modes, including sensitivity analysis of key parameters. The "Comparative Analysis under Different Cooperative Alliance Modes" section compares distinct decision-making approaches. The "Numerical Examples" section explores the effects of innovation realization rates and enhancement ratios on member strategies and performance, as well as the Pareto improvement effects of different cooperation methods on system profitability. Finally, the "Conclusion and Managerial Implications" section presents concluding thoughts and managerial implications.

## RELATED WORK

The article explores research areas such as green supply chain management, blockchain-enabled green supply chains, and optimizing investment strategies for green supply chain innovation.

Among the related studies on green supply chain management, *Jin (2021)* investigated the green supply chain management issues based on performance evaluation and considered the optimized green supply chain performance evaluation system for automobile enterprises with the goal of carbon peaking and carbon neutrality, highlighting the need for green and low-carbon development. *Kuiti et al. (2019)* analyzed the impact of green initiatives on profitability and waste reduction in a green supply chain. Numerous scholars have conducted research on the influence of consumers' environmental consciousness on green supply chain management. For instance, *Yao & Askin (2019)* concluded that the gradual rise in consumers' environmental awareness has a significant impact on companies' operational strategies in implementing green supply chain management. *Yalabik & Fairchild (2011)* found that consumers' environmental awareness positively motivates firms to invest in environmental innovation. This study enriches the existing research on green supply chain management by incorporating the innovation

factor from a dynamic perspective and utilizing the stochastic stopping problem to describe the phenomenon.

Blockchain-enabled green supply chains are gaining attention from both the business and academic communities, leading to research on various aspects of this subject. *Cao et al. (2022)* found that adding blockchain technology to traditional agricultural supply chains significantly improved yield and total surplus, positively incentivizing firms to invest in green initiatives. *Liu et al. (2020)* proposed a new supply chain structure to study investment decision-making and coordination in green agri-food supply chains using big data and blockchain. *Bai et al. (2021)* proposed a green data supply chain with a trust management mechanism to improve transparency and trust of the system. They evaluated its performance through simulations. It has been observed that most of the previous studies on blockchain-enabled green supply chains have focused on examining the effectiveness and impact of its implementation for businesses. However, they have overlooked the crucial decision-making issues that enterprises will face while applying blockchain technology. Therefore, this article takes into account the blockchain technology strategy developed by companies during their operations as a decision-making variable. The article investigates and analyses the issue of how blockchain technology strategy should be formulated for green supply chain members under the influence of innovation factors.

In terms of optimization of innovative investment strategies in green supply chains, *Tian, Li & Zhang (2022)* found that big data capabilities have a direct impact on green process innovation in the context of optimizing innovative investment strategies in green supply chains. The synergy of big data collection, analytics, and insights capabilities is crucial for achieving high levels of green process innovation. Academic research on innovation has been largely empirical in recent years, with both the business community and academics showing great interest in the topic (*Luu et al., 2023*; *Panda et al., 2023*; *Sørensen et al., 2023*; *Shah & Saqib, 2023*; *Jell-Ojobor, Alon & Windsperger, 2022*; *Hsu & Jang, 2009*; *Sadovnikova, Kacker & Mishra, 2023*). *Paparoidamis et al. (2019)* found that the degree of innovation in eco-friendly attributes positively affects consumers' perception of a product's eco-friendliness and willingness to adopt it. *Jiang, Mavondo & Zhao (2020)* found that network breadth and depth are important drivers of dynamic capabilities, which in turn facilitate successful product innovation in business networks. There is limited research available on product innovation in the theoretical community. However, this article will use the random stop problem to explain how uncertain the success time of product innovation can be. It will analyze how members of green supply chains can formulate their innovation investment strategy and select the cooperation mode from a dynamic perspective, considering the influence of various innovation factors.

A total of 162 journal articles on "blockchain-enabled green supply chain" were analyzed using CiteSpace software. The resulting knowledge map highlights current trends and hotspots. Figure 1 shows that green supply chain, agriculture, smart contracts, and blockchain technology are popular keywords.

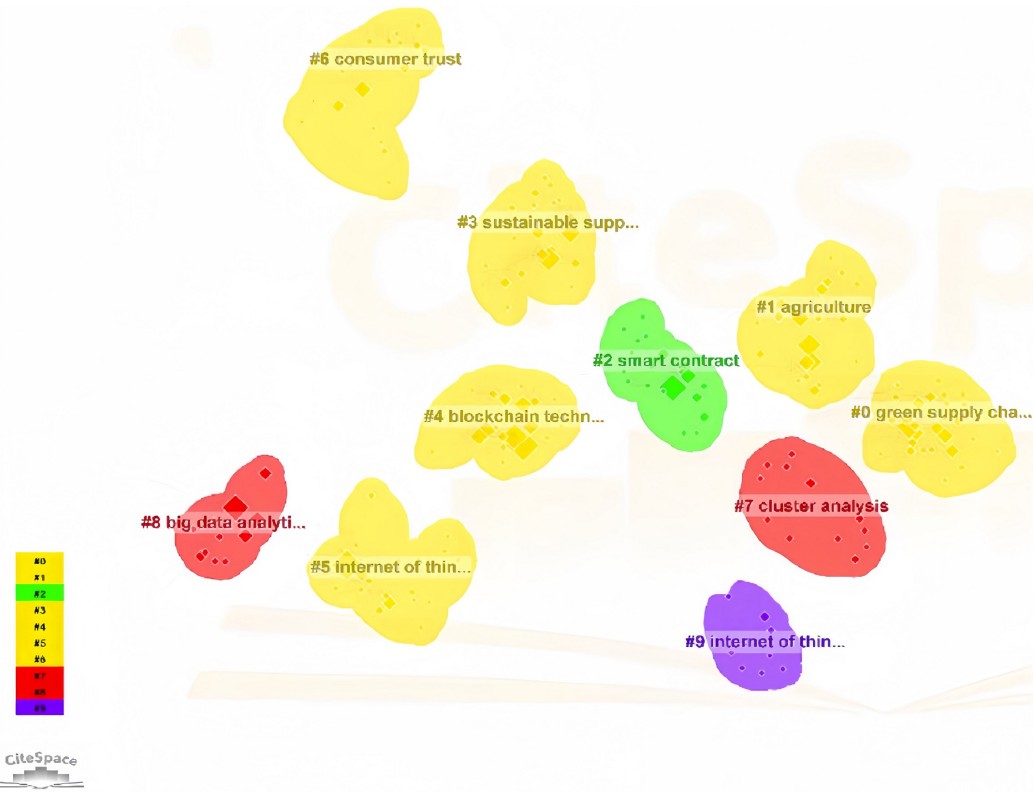

**Figure 1 Blockchain enabled green supply chain keywords cluster map.**

## PROBLEM DESCRIPTION

This article analyzes a green supply chain with a green supplier ($F$) and two Internet platforms ($M,N$) selling green products. The green supplier innovates to create environmentally-friendly products. The platforms prioritize eco-friendly transportation and packaging with sustainable materials. They invest in innovative strategies for green management and to promote environmental protection to more consumers (*Ju et al., 2023*; *Shou et al., 2023*). The platform will incorporate blockchain technology to tackle issues in green supply chain management, including information asymmetry, opacity, counterfeiting, and unethical practices. This will improve consumer trust and promote demand (*Liu et al., 2023*). In order to enhance consumer trust in green products, some platforms choose to use blockchain technology to establish a reliable traceability system for these products. However, this may inadvertently result in free-riding, where competitors also benefit from the increased sales generated by the improved consumer trust in green products. Despite this, the establishment of a credible green product traceability system can effectively promote the development of the industry and increase the desire of consumers to purchase these products (*Liu et al., 2023*). Some consumers may buy a green product from other platforms after receiving traceability information.

The concept of "green" has been gaining popularity in recent years, leading to the implementation of green supply chain management in various industries. Alongside this,

strategic alliances have become increasingly common in the green supply chain system. Strategic alliance cooperation typically involves both horizontal and vertical alliances (*Brookes & Roper, 2011*; *Gassenheimer, Baucus & Baucus, 1996*; *Boyle et al., 1992*; *Dant & Schul, 1992*). Horizontal alliance is when two platforms work together to decide on investment strategies and blockchain technology in order to limit competition and maximize joint profits. In contrast, vertical alliance involves a green supplier collaborating with one platform to jointly optimize profits, while the other platform makes independent decisions to maximize its own profits.

Product innovation is vital for modern businesses. With technology advancing rapidly, updates are frequent and the innovation cycle can be unpredictable (*Hao et al., 2019*). This article will study how green suppliers and internet platforms can optimize their operational strategies in response to product innovations in the green supply chain. It aims to improve resource allocation and achieve long-term and stable development. A system schematic of the green supply chain is shown in Fig. 2.

The decision-making sequence within the green supply chain starts with the green supplier developing the innovation investment strategy $a(t)$, and the two platforms decide on the innovation investment strategy $s(t)$ and blockchain technology $u(t)$, respectively. Green supply chain members who are forward-thinking understand that the introduction of new and innovative products can have a significant impact on the consumer market environment and the profits of all members involved. Therefore, they formulate operation strategies for two different environments: pre-innovation and post-innovation. They also consider the profits that can be achieved in these different environments before and after the innovation, in order to ensure that the overall profits are maximized. The green supplier will create two investment strategies, one for pre-innovation $a_1(t)$ and another for post-innovation $a_2(t)$. Similarly, the platform will also develop two investment strategies, one for pre-innovation $s_{x1}(t)$ and another for post-innovation $s_{x2}(t)$. The competitor platform $y$ will develop two strategies for investment in innovation, one for pre-innovation $s_{y1}(t)$ and another for post-innovation $s_{y2}(t)$, where $x, y \in \{M, N\}$ and $x \neq y$ denote Platform $M$ and Platform $N$, respectively. $i = 1, 2$ denotes two different environments before and after the success of product innovation, distinguishing that the evolution of green goodwill occurs in two different environments before and after the introduction of the innovative product. In addition, Platform $x$ will develop a pre-innovation blockchain technology level $u_{x1}(t)$ and a post-innovation blockchain technology level $u_{x2}(t)$, respectively. The decision timeline of the green supply chain is shown in Fig. 3. Table 1 summarizes the key symbols and detailed meanings in this article.

## MODEL ASSUPTION

**Hypothesis 1:** Having a good reputation is crucial for any successful product. This is particularly important in the context of green supply chain, where consumers rely heavily on accurate information related to product quality, eco-friendliness, and blockchain traceability. In fact, this information can be considered as the goodwill of the product in terms of its environmental impact (*Mushafiq, Prusak & Markiewicz, 2023*). There are numerous instances of innovation and improvement of eco-friendly products on online

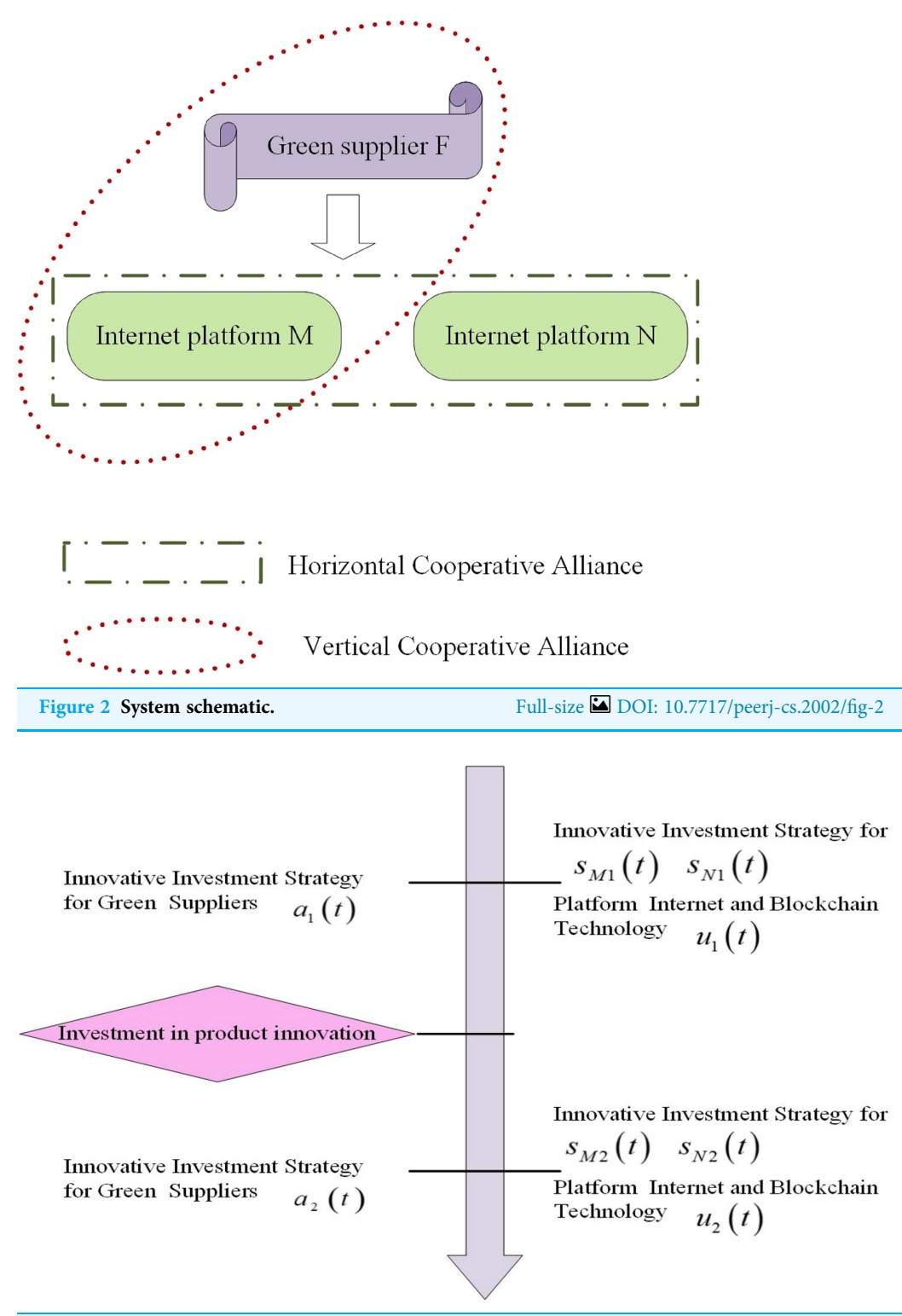

**Figure 2  System schematic.**               

**Figure 3  Decision-making timeline.**               

**Table 1 Key symbols and meaning.**

| Notation | Descriptions |
|---|---|
| **Parameter** | |
| $F, M, N$ | Green supply chain members, where $F$ represents green suppliers and $M$, $N$ represent two symmetric platforms respectively |
| $x, y$ | $x, y \in \{M, N\}$ and $x \neq y$ denote platform $M$ and platform $N$, respectively |
| $C, B, H$ | Possible scenarios representing horizontal cooperative alliances, non-cooperative models, vertical cooperative alliances |
| $i = 1, 2$ | Two different environments before and after the success of a product innovation distinguish the evolution of green goodwill as occurring in two different environments before and after the introduction of the innovative product |
| $\theta_i$ | Indicating the effectiveness of innovative investment strategies in enhancing green goodwill, $\theta_i > 0$ |
| $\alpha_i$ | The coefficient of impact of the platform's innovative investment strategy on green goodwill, $\alpha_i > 0$ |
| $\delta_i$ | Green goodwill decay factor, $\delta_i > 0$ |
| $\eta_i$ | Indicates the impact of platform x's innovative investment strategy on its own needs, $\eta_i > 0$ |
| $\xi_i$ | The impact of competitor y's innovative investment strategy on its own demand, using standard assumptions, $0 \leq \xi_i \leq \eta_i$ |
| $\lambda_i$ | the impact of the level of blockchain technology of platform x on its own needs, $\lambda_i > 0$ |
| $\gamma_i$ | positive spillover effects of competitor y's blockchain technology on platform x's needs, $\gamma_i > 0$ |
| $\omega_i > 0$ | Coefficient of impact of green goodwill on demand, $\omega_i > 0$ |
| $\rho_F, \rho_M, \rho_N$ | Margins for green suppliers and the two platforms, respectively |
| $k_F, k_{xu}, k_{xs}$ | Cost factors for green supplier innovation investments, platform innovation investments and blockchain technology, respectively, $k_F > 0, k_{xu} > 0, k_{xs} > 0$ |
| $r$ | discount rate |
| $\chi$ | Likelihood of success of product innovation, *i.e.*, realization rate |
| $\phi$ | Rate of increase in green goodwill due to successful product innovation |
| **Decision variables** | |
| $a_i(t)$ | Innovative investment strategies for green suppliers in the $i$ environment |
| $s_{xi}(t)$ | Innovative investment strategies for platform $x$ in an $i$ environment |
| $u_{xi}(t)$ | Level of blockchain technology for platform $x$ in $i$ environment |
| **State variables** | |
| $G(t)$ | Green Goodwill |

platforms to promote environmental consciousness. For instance, the E-commerce Innovation and Entrepreneurship Park situated in the Beilin District of Suihua City, Heilongjiang Province, China, is dedicated to implementing the "Internet +" sales model for agricultural products. Through this, it vigorously encourages the development of green agricultural products to create new markets and brands. The park continuously upgrades and optimizes its products to enhance their quality further. Suihua City's North Forest District established an official flagship store on Tmall to promote the unique characteristics of agricultural products. Qianmian E-commerce is selling agricultural and green products on internet platforms like Tmall, Alibaba, Suning, and Pinduoduo. These suppliers are implementing a green innovation investment strategy to create innovative green products using advanced green technologies and materials. Hewlett-Packard's suppliers are using green supply chain management and circular economy to achieve green product innovation and upgrading. This initiative can attract consumer attention, enhance

green goodwill, and maintain brand image (*Mushafiq, Prusak & Markiewicz, 2023*). The Internet platform has implemented a green innovation and investment strategy to achieve green management. This is done by improving and upgrading the logistics process, adopting environmentally friendly modes of transportation, optimizing and innovating packaging to improve energy efficiency in transportation, and implementing other green measures to create green logistics. They also use green packaging made of biodegradable and sustainable materials to promote the concept of environmental protection to more consumers. Green goodwill is not static and is always changing in a dynamic system. Therefore, green suppliers' R&D and innovation efforts, along with the platform's investment in a series of strategies to optimize green logistics and packaging, can drive the introduction of innovative products. This can help to build green goodwill across the entire green supply chain, benefitting the entire green supply chain system. It is assumed that the platforms are equally effective in improving goodwill (*Sigué, 2002*). In some cases, damage to goodwill can be caused by consumer forgetfulness or competition from other brands within the same industry (*He et al., 2018*; *Guo et al., 2021*). $G(t)$ is the stock of green goodwill. The kinetic equation is based on the (*Nerlove & Arrow, 1962*) goodwill model:

$$\dot{G}(t) = \theta_i \alpha_i(t) + \alpha_i s_{Mi}(t) + \alpha_i s_{Ni}(t) - \delta_i G, G(0) = G_0 > 0 \qquad (1)$$

**Hypothesis 2:** The National Development and Reform Commission and the Ministry of Commerce, along with other departments, are promoting the implementation of green logistics and packaging in the e-commerce, express delivery, and takeaway industries. This program requires e-commerce platforms, takeaway regulation platforms, and logistics companies to offer green consumption options. This includes promoting the standardization of e-commerce logistics, encouraging the use of recyclable and biodegradable packaging materials, and other sustainable logistics policies. Major platforms and enterprises are being encouraged to respond positively in order to establish the concept of green development and fulfill corporate social responsibility. For example, Suning has launched "zero rubber carton" packaging, and Jingdong uses recyclable red bags. Although these initiatives stimulate demand, they may also be negatively affected by competitors. However, by applying blockchain technology to realize anti-counterfeiting and traceability of green products, platforms can have a positive spillover effect and simultaneously promote demand for their own products and those of their competitors (*Liu et al., 2023*). In addition, investment in innovation by green suppliers can indirectly influence demand by accumulating goodwill which affects current and future sales, according to the interpretation of competition of *Liu et al. (2020)*. Therefore, the following demand function is given:

$$D_x = \eta_i s_{xi} - \xi_i s_{yi} + \lambda_i u_{xi} + \gamma_i u_{yi} + \omega_i G \qquad (2)$$

**Hypothesis 3:** To sharpen the focus of the study, the marginal profit $\rho_F$, $\rho_M$, $\rho_N$ of both green suppliers and platforms is assumed to be constant, while concentrating on strategies such as innovative investment strategies and blockchain technology (*Chintagunta & Jain, 1992*). Our model makes the assumption of setting the marginal profit as a fixed parameter.

This is done to simplify the model and to focus on analyzing the optimization study of innovative investment strategies and blockchain technology strategies. However, we acknowledge that this assumption is a limitation of our model.

**Hypothesis 4:** In order to innovate a product, it takes a certain amount of time. It is also uncertain when the innovation will succeed (*Hao et al., 2019*). The occurrence of green product innovation and the research and development process can be represented by $\{\Gamma(t) : t > 0\}$. The probability of success of the innovation at any moment $t$ is $\chi \in (0, 1)$. Assuming $T$ represents the date when innovation success was achieved, the operation time of the green supply chain can be divided into two intervals: pre-innovation $t \in [0, T]$ and post-innovation $t \in [T, +\infty]$. Referring to the works of *Van Heerde, Helsen & Dekimpe (2007)* and *Rubel, Naik & Srinivasan (2011)*, $\{\Gamma(t) : t > 0\}$ represents a jumping process that can be expressed as a mathematical function

$$\lim_{\Delta \to 0} \frac{P[\Gamma(t + \Delta) = 2 \mid \Gamma(t) = 1]}{\Delta} = \chi, \lim_{\Delta \to 0} \frac{P[\Gamma(t) = 1 \mid \Gamma(t + \Delta) = 2]}{\Delta} = 0 \tag{3}$$

when companies introduce innovative products, the public's reaction to them can have a big impact on the company's reputation. For example, in September of last year, Starbucks collaborated with artist Steven Harrington to release a limited number of Mason Jars decorated with pop-style illustrations. These cute and practical cups were highly sought-after by young people in the US, and this partnership helped boost Starbucks' reputation. The introduction of innovative products can lead to an increase in goodwill, as noted by *Van Heerde, Helsen & Dekimpe (2007)* and *Rubel, Naik & Srinivasan (2011)*. This process can be described as

$$G(T^+) = (1 + \phi)G(T^-), G(T^+) \neq G(T^-) \tag{4}$$

**Hypothesis 5:** Assuming a quadratic cost function, as commonly used in previous studies (*Martín-Herrán, Sigué & Zaccour, 2011*), the innovation investment cost for the green supplier is $k_F a^2(t)/2$, The innovation investment cost for the platform is $k_{xu} u_x^2(t)/2$, while the cost of the blockchain technology is $k_{xs} s_x^2(t)/2$. $x = M, N$ denotes Platform $M$ and Platform $N$, respectively. $k_{xu} > 0$, $k_{xs} > 0$ denote the respective cost coefficients.

**Hypothesis 6:** The successful launch of innovative products will make the green supply chain different in the pre-innovation and post-innovation environments, and the visionary green supply chain members will consider the operation strategies and profits under the pre-innovation and post-innovation phases, so as to formulate the optimal operation strategies under the different phases, and rationally allocate the resources in order to achieve the maximization of the overall profits. Since the enterprise does not know the exact time of innovation realization $T$, with $\pi_i$ before and after the innovation of profit, the net present value of its profits, respectively, $J_1 = \int_0^T e^{-rt} \pi_1 dt$, $J_2 = \int_T^\infty e^{-rt} \pi_2 dt$, two environments of the expected net present value of profits for the $J = E(J_1 + e^{-rt} J_2)$, the simplification of the collation to obtain the long-term profit function

$$J = \int_0^\infty e^{-(r+\chi)} \{\pi_1 + \chi J_2\} dt \tag{5}$$

## MODEL ANALYSIS

This section will solve each strategy and profit of green supply chain members under the three modes of horizontal cooperative alliance (C), non-cooperative (B), and vertical cooperative alliance (H), respectively. It will also carry out the sensitivity analysis of the key parameters and provide managerial insights accordingly. To differentiate the models, the superscripts *C*, *B*, and *H* are used to denote the three modes, and the subscripts *F*, *M*, and *N* are used to denote the green suppliers and the two platforms, respectively. Proofs of all propositions and properties in the text can be found in the Supplemental Articles.

### Horizontal cooperative alliance model (*C*)

In the context of a horizontal cooperative alliance model, platforms work together to reduce competition in local markets and jointly make decisions on investing in blockchain technology. Their ultimate goal is to maximize their collective profits. Decision-makers approach this model strategically by considering the potential impact of future product innovations on the overall system. We denote this model as *C* through a superscript. The differential game model for this scenario can be summarized as follows:

$$max\left\{ J_F = \int_0^\infty e^{-rt}\left[ \rho_{F1}(D_{M1}(t) + D_{N1}(t)) - \frac{1}{2}k_F a_1{}^2(t) + \chi W_F^C((1+\phi)G(t)) \right]dt \right\}$$

$$max\left\{ J_{MN} = \int_0^\infty e^{-rt}\begin{bmatrix} \rho_{M1}D_{M1}(t) + \rho_{N1}D_{N1}(t) - \frac{1}{2}k_{Mu}u_{M1}^2(t) - \frac{1}{2}k_{Ms}s_{M1}^2(t) \\ -\frac{1}{2}k_{Nu}u_{N1}^2(t) - \frac{1}{2}k_{Ns}s_{N1}^2(t) + \chi W_{MN}^C((1+\phi)G(t)) \end{bmatrix}dt \right\} \tag{6}$$

**Proposition 1:** (1) The innovation investment strategies of pre-innovation and post-innovation green suppliers are respectively

$$a_1^C = \frac{\theta_1 d_4}{k_F}; a_2^C = \frac{\theta_2 d_{11}}{k_F} \tag{7}$$

The innovative investment strategies and the level of blockchain technology of the pre-innovation and post-innovation collaborative platforms are respectively

$$s_{x1}^C = \frac{\rho_{x1}\eta_1 - \rho_{y1}\xi_1 + \alpha_1 d_3}{k_{xs}}; u_{x1}^C = \frac{\rho_{x1}\lambda_1 + \rho_{y1}\gamma_1}{k_{xu}}; s_{x2}^C = \frac{\rho_{x2}\eta_2 - \rho_{y2}\xi_2 + \alpha_2 d_{22}}{k_{xs}};$$

$$u_{x2}^C = \frac{\rho_{x2}\lambda_2 + \rho_{y2}\gamma_2}{k_{xu}} \tag{8}$$

(2) The time evolution path of green goodwill is

$$G^C(t) = \begin{cases} G_{1\infty}^C + (G_0 - G_{1\infty}^C)e^{-\delta_1 t} & t \in [0, T] \\ G_{2\infty}^C + [(1+\phi)G_T^C - G_{2\infty}^C]e^{-\delta_2(t-T)} & t \in [T, \infty) \end{cases} \tag{9}$$

Among them,

$$G_T^C = G_{1\infty}^C + (G_0 - G_{1\infty}^C)e^{-\delta_1 T} \tag{10}$$

$$G_{1\infty}^C = \frac{1}{\delta_1}\left[\frac{\theta_1^2 d_4}{k_F} + \frac{\alpha_1(\rho_{M1}\eta_1 - \rho_{N1}\xi_1 + \alpha_1 d_3)}{k_{Ms}} + \frac{\alpha_1(\rho_{N1}\eta_1 - \rho_{M1}\xi_1 + \alpha_1 d_3)}{k_{Ns}}\right] \tag{11}$$

$$G_{2\infty}^C = \frac{1}{\delta_2}\left[\frac{\theta_2^2 d_{11}}{k_F} + \frac{\alpha_2(\rho_{M2}\eta_2 - \rho_{N2}\xi_2 + \alpha_2 d_{22})}{k_{Ms}} + \frac{\alpha_2(\rho_{N2}\eta_2 - \rho_{M2}\xi_2 + \alpha_2 d_{22})}{k_{Ns}}\right] \tag{12}$$

(3) The profits of green suppliers and cooperative platforms for the entire program period are respectively

$$V_F^C = d_4 G + d_{04}; \quad V_{MN}^C = d_3 G + d_{03} \tag{13}$$

The profits of the green supplier and the collaborative platform after the innovation are respectively

$$W_F^C = d_{11}G + d_{01}; \quad W_{MN}^C = d_{22}G + d_{02} \tag{14}$$

Among them,

$$d_4 = \frac{2\rho_{F1}\omega_1 + \chi d_{11}(1+\phi)}{r+\chi+\delta_1}; \quad d_3 = \frac{\rho_{M1}\omega_1 + \rho_{N1}\omega_1 + \chi d_{22}(1+\phi)}{r+\chi+\delta_1}; \quad d_{11} = \frac{2\rho_{F2}\omega_2}{r+\delta_2};$$
$$d_{22} = \frac{(\rho_{M2} + \rho_{N2})\omega_2}{r+\delta_2} \tag{15}$$

$$d_{04} = \frac{1}{r+\chi}\begin{bmatrix} \dfrac{(\rho_{F1}\eta_1 + \alpha_1 d_4 - \rho_{F1}\xi_1)(\rho_{M1}\eta_1 - \xi_1\rho_{N1} + \alpha_1 d_3)}{k_{Ms}} \\[2mm] + \dfrac{(\rho_{F1}\lambda_1 + \rho_{F1}\gamma_1)(\rho_{M1}\lambda_1 + \rho_{N1}\gamma_1)}{k_{Mu}} + \dfrac{(\theta_1 d_4)^2}{2k_F} \\[2mm] + \dfrac{(\rho_{F1}\eta_1 + \alpha_1 d_4 - \rho_{F1}\xi_1)(\rho_{N1}\eta_1 - \xi_1\rho_{M1} + \alpha_1 d_3)}{k_{Ns}} \\[2mm] + \dfrac{(\rho_{F1}\gamma_1 + \rho_{F1}\lambda_1)(\rho_{M1}\gamma_1 + \rho_{N1}\lambda_1)}{k_{Nu}} + \chi d_{01} \end{bmatrix} \tag{16}$$

$$d_{03} = \frac{1}{r+\chi}\begin{bmatrix} \dfrac{(\rho_{M1}\eta_1 - \xi_1\rho_{N1} + \alpha_1 d_3)^2}{2k_{Ms}} + \dfrac{(\rho_{M1}\lambda_1 + \rho_{N1}\gamma_1)^2}{2k_{Mu}} \\[2mm] + \dfrac{(\eta_1\rho_{N1} - \xi_1\rho_{M1} + \alpha_1 d_3)^2}{2k_{Ns}} + \dfrac{(\rho_{M1}\gamma_1 + \rho_{N1}\lambda_1)^2}{2k_{Nu}} \\[2mm] + \dfrac{\theta_1^2 d_3 d_4}{k_F} + \chi d_{02} \end{bmatrix} \tag{17}$$

$$d_{01} = \frac{1}{r}\begin{bmatrix} \dfrac{(\rho_{F2}\eta_2 + \alpha_2 d_{11} - \rho_{F2}\xi_2)(\rho_{M2}\eta_2 - \rho_{N2}\xi_2 + \alpha_2 d_{22})}{k_{Ms}} \\[2mm] + \dfrac{(\rho_{F2}\lambda_2 + \rho_{F2}\gamma_2)(\rho_{M2}\lambda_2 + \rho_{N2}\gamma_2)}{k_{Mu}} \\[2mm] + \dfrac{(\rho_{F2}\eta_2 + \alpha_2 d_{11} - \rho_{F2}\xi_2)(\rho_{N2}\eta_2 - \rho_{M2}\xi_2 + \alpha_2 d_{22})}{k_{Ns}} \\[2mm] + \dfrac{(\rho_{F2}\lambda_2 + \rho_{F2}\gamma_2)(\rho_{M2}\gamma_2 + \rho_{N2}\lambda_2)}{k_{Nu}} + \dfrac{(\theta_2 d_{11})^2}{2k_F} \end{bmatrix} \tag{18}$$

$$d_{02} = \frac{1}{r} \left[ \begin{array}{c} \dfrac{(\rho_{M2}\eta_2 - \rho_{N2}\xi_2 + \alpha_2 d_{22})^2}{2k_{Ms}} + \dfrac{(\rho_{M2}\lambda_2 + \rho_{N2}\gamma_2)^2}{2k_{Mu}} \\[3mm] + \dfrac{(\rho_{N2}\eta_2 - \rho_{M2}\xi_2 + \alpha_2 d_{22})^2}{2k_{Ns}} + \dfrac{(\rho_{M2}\gamma_2 + \rho_{N2}\lambda_2)^2}{2k_{Nu}} + \dfrac{\theta_2^2 d_{11} d_{22}}{k_F} \end{array} \right] \tag{19}$$

## Non-cooperative modalities (*B*)

In the non-cooperative model, the three enterprises in the green supply chain system function as independent economic entities, each pursuing their own profit goals. To achieve maximum profits over the planning horizon, decision-makers must identify and implement optimal strategies before and after innovation. The superscript *B* indicates the non-cooperative mode. At this point, the differential game model can be expressed as follows:

$$max\left\{ J_F = \int_0^\infty e^{-rt} \left[ \rho_{F1}(D_{M1}(t) + D_{N1}(t)) - \frac{1}{2}k_F a_1^2(t) + \chi W_F^B((1+\phi)G(t)) \right] dt \right\}$$

$$max\left\{ J_M = \int_0^\infty e^{-rt} \left[ \rho_{M1}D_{M1}(t) - \frac{1}{2}k_{Mu}u_{M1}^2(t) - \frac{1}{2}k_{Ms}s_{M1}^2(t) + \chi W_M^B((1+\phi)G(t)) \right] dt \right\} \tag{20}$$

$$max\left\{ J_N = \int_0^\infty e^{-rt} \left[ \rho_{N1}D_{N1}(t) - \frac{1}{2}k_{Nu}u_{N1}^2(t) - \frac{1}{2}k_{Ns}s_{N1}^2(t) + \chi W_N^B((1+\phi)G(t)) \right] dt \right\}$$

**Proposition 2:** (1) The innovation investment strategies of pre-innovation and post-innovation green suppliers are respectively

$$a_1^B = \frac{\theta_1 d_8}{k_F}; a_2^B = \frac{\theta_2 d_5}{k_F} \tag{21}$$

The innovative investment strategies and the level of blockchain technology of each platform before and after the innovation are as follows

$$s_{x1}^B = \frac{\rho_{x1}\eta_1 + \alpha_1 d_9}{k_{xs}}; u_{x1}^B = \frac{\rho_{x1}\lambda_1}{k_{xu}}; s_{x2}^B = \frac{\rho_{x2}\eta_2 + \alpha_2 d_6}{k_{xs}}; u_{x2}^B = \frac{\rho_{x2}\lambda_2}{k_{xu}} \tag{22}$$

(2) The time evolution path of green goodwill is

$$G^B(t) = \begin{cases} G_{1\infty}^B + (G_0 - G_{1\infty}^B)e^{-\delta_1 t} & t \in [0, T] \\ G_{2\infty}^B + [(1+\phi)G_T^B - G_{2\infty}^B]e^{-\delta_2(t-T)} & t \in [T, \infty) \end{cases} \tag{23}$$

$$G_T^B = G_{1\infty}^B + (G_0 - G_{1\infty}^B)e^{-\delta_1 T} \tag{24}$$

$$G_{1\infty}^B = \frac{1}{\delta_1} \left[ \frac{\theta_1^2 d_8}{k_F} + \frac{\alpha_1(\rho_{M1}\eta_1 + \alpha_1 d_9)}{k_{Ms}} + \frac{\alpha_1(\eta_1 \rho_{N1} + \alpha_1 d_{10})}{k_{Ns}} \right] \tag{25}$$

$$G_{2\infty}^B = \frac{1}{\delta_2} \left[ \frac{\theta_2^2 d_5}{k_F} + \frac{\alpha_2(\rho_{M2}\eta_2 + \alpha_2 d_6)}{k_{Ms}} + \frac{\alpha_2(\rho_{N2}\eta_2 + \alpha_2 d_7)}{k_{Ns}} \right] \tag{26}$$

(3) The profits of green suppliers and platforms for the entire program period are respectively

$$V_F^B = d_8 G + d_{08}; \quad V_M^B = d_9 G + d_{09}; \quad V_N^B = d_{10} G + d_{010} \tag{27}$$

The profits of the green supplier and the collaborative platform after the innovation are respectively

$$W_F^B = d_5 G + d_{05}; \quad W_M^B = d_6 G + d_{06}; \quad W_N^B = d_7 G + d_{07} \tag{28}$$

Among them,

$$d_8 = \frac{2\omega_1 \rho_{F1} + \chi d_5(1+\phi)}{r + \chi + \delta_1}; \quad d_9 = \frac{\rho_{M1}\omega_1 + \chi d_6(1+\phi)}{r + \chi + \delta_1}; \quad d_{10} = \frac{\rho_{N1}\omega_1 + \chi d_7(1+\phi)}{r + \chi + \delta_1};$$
$$d_5 = \frac{2\omega_2 \rho_{F2}}{r + \delta_2}; \quad d_6 = \frac{\rho_{M2}\omega_2}{r + \delta_2}; \quad d_7 = \frac{\rho_{N2}\omega_2}{r + \delta_2} \tag{29}$$

$$d_{08} = \frac{1}{r+\chi} \begin{bmatrix} \dfrac{(\rho_{F1}\eta_1 + d_8\alpha_1 - \rho_{F1}\xi_1)(\rho_{M1}\eta_1 + \alpha_1 d_9)}{k_{Ms}} \\[2ex] + \dfrac{(\rho_{F1}\eta_1 + \alpha_1 d_8 - \rho_{F1}\xi_1)(\eta_1\rho_{N1} + \alpha_1 d_{10})}{k_{Ns}} \\[2ex] + \dfrac{(\rho_{F1}\lambda_1 + \rho_{F1}\gamma_1)\rho_{M1}\lambda_1}{k_{Mu}} + \dfrac{(\rho_{F1}\gamma_1 + \rho_{F1}\lambda_1)\rho_{N1}\lambda_1}{k_{Nu}} \\[2ex] + \dfrac{(\theta_1 d_8)^2}{2k_F} + \chi d_{05} \end{bmatrix} \tag{30}$$

$$d_{09} = \frac{1}{r+\chi} \begin{bmatrix} \dfrac{(\rho_{M1}\eta_1 + \alpha_1 d_9)^2}{2k_{Ms}} + \dfrac{(\alpha_1 d_9 - \rho_{M1}\xi_1)(\eta_1\rho_{N1} + \alpha_1 d_{10})}{k_{Ns}} \\[2ex] + \dfrac{(\rho_{M1}\lambda_1)^2}{2k_{Mu}} + \dfrac{\rho_{M1}\rho_{N1}\gamma_1\lambda_1}{k_{Nu}} + \dfrac{\theta_1^2 d_9 d_8}{k_F} + \chi d_{06} \end{bmatrix} \tag{31}$$

$$d_{010} = \frac{1}{r+\chi} \begin{bmatrix} \dfrac{(\eta_1\rho_{N1} + \alpha_1 d_{10})^2}{2k_{Ns}} + \dfrac{(d_{10}\alpha_1 - \rho_{N1}\xi_1)(\rho_{M1}\eta_1 + \alpha_1 d_9)}{k_{Ms}} \\[2ex] + \dfrac{(\lambda_1\rho_{N1})^2}{2k_{Nu}} + \dfrac{\rho_{N1}\rho_{M1}\gamma_1\lambda_1}{k_{Mu}} + \dfrac{d_{10}d_8\theta_1^2}{k_F} + \chi d_{07} \end{bmatrix} \tag{32}$$

$$d_{05} = \frac{1}{r} \begin{bmatrix} \dfrac{(\rho_{F2}\eta_2 + \alpha_2 d_5 - \rho_{F2}\xi_2)(\rho_{M2}\eta_2 + \alpha_2 d_6)}{k_{Ms}} \\[2ex] + \dfrac{(\rho_{F2}\eta_2 + \alpha_2 d_5 - \rho_{F2}\xi_2)(\rho_{N2}\eta_2 + \alpha_2 d_7)}{k_{Ns}} \\[2ex] + \dfrac{(\rho_{F2}\gamma_2 + \rho_{F2}\lambda_2)\rho_{M2}\lambda_2}{k_{Mu}} + \dfrac{(\rho_{F2}\gamma_2 + \rho_{F2}\lambda_2)\rho_{N2}\lambda_2}{k_{Nu}} + \dfrac{(\theta_2 d_5)^2}{2k_F} \end{bmatrix} \tag{33}$$

$$d_{06} = \frac{1}{r} \begin{bmatrix} \dfrac{(\rho_{M2}\eta_2 + \alpha_2 d_6)^2}{2k_{Ms}} + \dfrac{(\alpha_2 d_6 - \rho_{M2}\xi_2)(\rho_{N2}\eta_2 + \alpha_2 d_7)}{k_{Ns}} \\[2ex] + \dfrac{(\rho_{M2}\lambda_2)^2}{2k_{Mu}} + \dfrac{\rho_{N2}\rho_{M2}\gamma_2\lambda_2}{k_{Nu}} + \dfrac{d_6 d_5 \theta_2^2}{k_F} \end{bmatrix} \tag{34}$$

$$d_{07} = \frac{1}{r}\left[\begin{array}{l}\dfrac{(\rho_{N2}\eta_2 + \alpha_2 d_7)^2}{2k_{Ns}} + \dfrac{(d_7\alpha_2 - \rho_{N2}\xi_2)(\rho_{M2}\eta_2 + \alpha_2 d_6)}{k_{Ms}} \\[2mm] + \dfrac{(\rho_{N2}\lambda_2)^2}{2k_{Nu}} + \dfrac{\rho_{N2}\gamma_2\rho_{M2}\lambda_2}{k_{Mu}} + \dfrac{d_7 d_5 \theta_2^2}{k_F}\end{array}\right] \tag{35}$$

## Vertical cooperative alliance model (*H*)

Within this model, the green supplier enters into a cooperative partnership with one of the platforms, working in unison to determine the optimal strategy aimed at maximizing their collective profit. Concurrently, the other platform acts independently, strategizing to enhance its own profitability. Each decision-maker, cognizant of the potential repercussions of innovation, meticulously devises strategies to accommodate both the pre-innovation and post-innovation stages. The superscript *H* denotes the vertical cooperative alliance model, under which the differential game model can be thusly articulated:

$$max\left\{J_{FM} = \int_0^\infty e^{-rt}\left[\begin{array}{l}\rho_{F1}(D_{M1}(t) + D_{N1}(t)) + \rho_{M1}D_{M1}(t) - \frac{1}{2}k_F a_1^2(t) - \frac{1}{2}k_{Mu}u_{M1}^2(t) \\[2mm] -\frac{1}{2}k_{Ms}s_{M1}^2(t) + \chi W_{FM}^H((1+\phi)G(t))\end{array}\right]dt\right\}$$

$$max\left\{J_N = \int_0^\infty e^{-rt}\left[\rho_{N1}D_{N1}(t) - \frac{1}{2}k_{Nu}u_{N1}^2(t) - \frac{1}{2}k_{Ns}s_{N1}^2(t) + \chi W_N^H((1+\phi)G(t))\right]dt\right\} \tag{36}$$

**Proposition 3:** (1) The innovation investment strategies of pre-innovation and post-innovation green suppliers are respectively

$$a_1^H = \frac{\theta_1 d_{14}}{k_F}; a_2^H = \frac{\theta_2 d_{12}}{k_F} \tag{37}$$

The innovative investment strategies and the level of blockchain technology of each platform before and after the innovation are as follows

$$s_{M1}^H = \frac{\rho_{F1}(\eta_1 - \xi_1) + \rho_{M1}\eta_1 + \alpha_1 d_{14}}{k_{Ms}}; s_{N1}^H = \frac{\rho_{N1}\eta_1 + \alpha_1 d_{15}}{k_{Ns}};$$

$$u_{M1}^H = \frac{\rho_{F1}(\lambda_1 + \gamma_1) + \rho_{M1}\lambda_1}{k_{Mu}}; u_{N1}^H = \frac{\rho_{N1}\lambda_1}{k_{Nu}};$$

$$s_{M2}^H = \frac{\rho_{F2}(\eta_2 - \xi_2) + \rho_{M2}\eta_2 + d_{12}\alpha_2}{k_{Ms}}; s_{N2}^H = \frac{\rho_{N2}\eta_2 + \alpha_2 d_{13}}{k_{Ns}};$$

$$u_{M2}^H = \frac{\rho_{F2}(\gamma_2 + \lambda_2) + \rho_{M2}\lambda_2}{k_{Mu}}; u_{N2}^H = \frac{\rho_{N2}\lambda_2}{k_{Nu}} \tag{38}$$

(2) The time evolution path of green goodwill is

$$G^H(t) = \begin{cases} G_{1\infty}^H + (G_0 - G_{1\infty}^H)e^{-\delta_1 t} & t \in [0, T] \\[2mm] G_{2\infty}^H + [(1+\phi)G_T^H - G_{2\infty}^H]e^{-\delta_2(t-T)} & t \in [T, \infty) \end{cases} \tag{39}$$

Among them,

$$G_T^H = G_{1\infty}^H + (G_0 - G_{1\infty}^H)e^{-\delta_1 t} \tag{40}$$

$$G_{1\infty}^H = \frac{1}{\delta_1}\left[\frac{\theta_1^2 d_{14}}{k_F} + \frac{\alpha_1(\rho_{F1}(\eta_1 - \xi_1) + \rho_{M1}\eta_1 + \alpha_1 d_{14})}{k_{Ms}} + \frac{\alpha_1(\rho_{N1}\eta_1 + \alpha_1 d_{15})}{k_{Ns}}\right] \tag{41}$$

$$G_{2\infty}^H = \frac{1}{\delta_2}\left[\frac{\theta_2^2 d_{12}}{k_F} + \frac{\alpha_2(\rho_{F2}(\eta_2 - \xi_2) + \rho_{M2}\eta_2 + d_{12}\alpha_2)}{k_{Ms}} + \frac{\alpha_2(\rho_{N2}\eta_2 + \alpha_2 d_{13})}{k_{Ns}}\right] \tag{42}$$

(3) The profits of cooperating green suppliers and platforms and independent platforms for the entire program period are respectively

$$V_{FM}^H = d_{14}G + d_{014}; V_N^H = d_{15}G + d_{015} \tag{43}$$

The profits of the post-innovation cooperative green suppliers and platforms, and independent platforms, respectively, were

$$W_{FM}^H = d_{12}G + d_{012}; W_N^H = d_{13}G + d_{013} \tag{44}$$

Among them,

$$d_{14} = \frac{(2\rho_{F1} + \rho_{M1})\omega_1 + \chi d_{12}(1 + \phi)}{r + \chi + \delta_1}; d_{15} = \frac{\rho_{N1}\omega_1 + \chi d_{13}(1 + \phi)}{r + \chi + \delta_1};$$
$$d_{12} = \frac{(2\rho_{F2} + \rho_{M2})\omega_2}{r + \delta_2}; d_{13} = \frac{\rho_{N2}\omega_2}{r + \delta_2} \tag{45}$$

$$d_{014} = \frac{1}{r + \chi}\left[\begin{array}{l} \dfrac{(\rho_{F1}(\eta_1 - \xi_1) + \rho_{M1}\eta_1 + \alpha_1 d_{14})^2}{2k_{Ms}} \\[2mm] + \dfrac{(\rho_{F1}(\eta_1 - \xi_1) - \rho_{M1}\xi_1 + \alpha_1 d_{14})(\rho_{N1}\eta_1 + \alpha_1 d_{15})}{k_{Ns}} \\[2mm] + \dfrac{(\rho_{F1}(\lambda_1 + \gamma_1) + \rho_{M1}\lambda_1)^2}{2k_{Mu}} \\[2mm] + \dfrac{(\rho_{F1}(\gamma_1 + \lambda_1) + \rho_{M1}\gamma_1)\rho_{N1}\lambda_1}{k_{Nu}} + \dfrac{(\theta_1 d_{14})^2}{k_F} + \chi d_{012} \end{array}\right] \tag{46}$$

$$d_{015} = \frac{1}{r + \chi}\left[\begin{array}{l} \dfrac{(\rho_{N1}\eta_1 + \alpha_1 d_{15})^2}{2k_{Ns}} \\[2mm] + \dfrac{(d_{15}\alpha_1 - \rho_{N1}\xi_1)(\rho_{F1}(\eta_1 - \xi_1) + \rho_{M1}\eta_1 + \alpha_1 d_{14})}{k_{Ms}} + \dfrac{(\rho_{N1}\lambda_1)^2}{2k_{Nu}} \\[2mm] + \dfrac{\rho_{N1}\gamma_1(\rho_{F1}(\lambda_1 + \gamma_1) + \rho_{M1}\lambda_1)}{k_{Mu}} + \dfrac{d_{15}d_{14}\theta_1^2}{k_F} + \chi d_{013} \end{array}\right] \tag{47}$$

$$d_{012} = \frac{1}{r}\left[\begin{array}{l} \dfrac{(\rho_{F2}(\eta_2 - \xi_2) + \rho_{M2}\eta_2 + \alpha_2 d_{12})^2}{2k_{Ms}} \\[2mm] + \dfrac{(\rho_{F2}(\eta_2 - \xi_2) - \rho_{M2}\xi_2 + \alpha_2 d_{12})(\rho_{N2}\eta_2 + \alpha_2 d_{12})}{k_{Ns}} \\[2mm] + \dfrac{(\rho_{F2}(\gamma_2 + \lambda_2) + \rho_{M2}\lambda_2)^2}{2k_{Mu}} + \dfrac{(\rho_{F2}(\gamma_2 + \lambda_2) + \rho_{M2}\gamma_2)\rho_{N2}\lambda_2}{k_{Nu}} + \dfrac{(\theta_2 d_{12})^2}{2k_F} \end{array}\right] \tag{48}$$

$$d_{013} = \frac{1}{r} \left[ \begin{array}{c} \dfrac{(\rho_{N2}\eta_2 + \alpha_2 d_{13})^2}{2k_{Ns}} + \dfrac{(d_{13}\alpha_2 - \rho_{N2}\xi_2)(\rho_{F2}(\eta_2 - \xi_2) + \rho_{M2}\eta_2 + \alpha_2 d_{12})}{k_{Ms}} \\ + \dfrac{(\rho_{N2}\lambda_2)^2}{2k_{Nu}} + \dfrac{\rho_{N2}\gamma_2(\rho_{F2}(\gamma_2 + \lambda_2) + \rho_{M2}\lambda_2)}{k_{Mu}} + \dfrac{\theta_2^2 d_{12} d_{13}}{k_F} \end{array} \right] \tag{49}$$

**Proposition 4:** Aside from the decision variables, the text also includes fixed parameters known as exogenous parameters. As such, Tables 2–4 displays the sensitivity analysis of the green suppliers' innovation investment strategy and cooperative platforms' innovation investment strategy with regards to the key exogenous parameters under the three models, both pre- and post-innovation.

Proposition 4 suggests that if companies work together in a green supply chain, investing in innovation to improve eco-friendliness and sustainability can have a positive impact on their overall green reputation. This encourages suppliers to increase their efforts in research and development, leading to the integration of advanced green technologies and eco-friendly materials in their products. The companies in the green supply chain are also aware of the importance of green reputation and how it can affect demand at all levels. This encourages suppliers to refine their innovation investment strategies before introducing new products. In a horizontal cooperative alliance, companies work together and consider the impact of their innovation investment strategies on each other's demand. When a company's innovation investment strategy negatively affects its partner's demand, it reduces its own innovation investment strategy to maintain profitability. This is expected as competition is reduced when there is cooperation. In a non-cooperative model, competition between companies is higher. The results of the sensitivity analysis of the innovation investment strategies of green suppliers and platforms before and after innovation are similar to Proposition 2. However, in this model, the platform's innovation investment strategy is not affected by the competition's innovation investment strategy. In a vertical alliance model, the platform's innovation investment strategy is positively correlated with its own demand and negatively correlated with the impact of the competition's innovation investment strategy on its own demand. In this context, the platform's innovation investment strategy is only positively correlated with its own demand.

**Proposition 5:** The sensitivity analysis of the level of blockchain technology regarding the key parameters in the horizontal cooperative alliance model are $\frac{\partial u_{x1}^C}{\partial \lambda_1} > 0$, $\frac{\partial u_{x1}^C}{\partial \gamma_1} > 0$, $\frac{\partial u_{x2}^C}{\partial \lambda_2} > 0$, $\frac{\partial u_{x2}^C}{\partial \gamma_2} > 0$, respectively.

According to Proposition 5, a platform's level of blockchain technology is not only linked to its own influence on demand, but also to its partner's blockchain technology level and its influence on the platform's demand. This highlights the importance of considering both factors when forming horizontal cooperative alliances on collaborative platforms.

**Proposition 6:** The sensitivity analysis of the level of blockchain technology regarding the key parameters in the non-cooperative mode are $\frac{\partial u_{x1}^B}{\partial \lambda_1} > 0$, $\frac{\partial u_{x1}^B}{\partial \gamma_1} = 0$, $\frac{\partial u_{x2}^B}{\partial \lambda_2} > 0$, $\frac{\partial u_{x2}^B}{\partial \gamma_2} = 0$ respectively.

**Table 2 Sensitivity analysis of exogenous parameters under the horizontal cooperative alliance model.**

| | $\theta_1$ | $\theta_2$ | $\alpha_1$ | $\alpha_2$ | $\delta_1$ | $\delta_2$ | $\eta_1$ | $\eta_2$ | $\xi_1$ | $\xi_2$ | $\omega_1$ | $\omega_2$ |
|---|---|---|---|---|---|---|---|---|---|---|---|---|
| $a_1^C$ | ↗ | — | — | — | ↘ | ↘ | — | — | — | — | ↗ | ↗ |
| $a_2^C$ | — | ↗ | — | — | — | ↘ | — | — | — | — | — | ↗ |
| $s_{x1}^C$ | — | — | ↗ | — | ↘ | ↘ | ↗ | — | ↘ | — | ↗ | ↗ |
| $s_{x2}^C$ | — | — | — | ↗ | — | ↗ | — | ↘ | — | ↘ | — | ↗ |

Note:
↗ indicates a positive correlation with exogenous parameters, ↘ indicates a negative correlation with exogenous parameters, and the horizontal line indicates no correlation with exogenous parameters.

**Table 3 Sensitivity analysis of exogenous parameters in non-cooperative mode.**

| | $\theta_1$ | $\theta_2$ | $\alpha_1$ | $\alpha_2$ | $\delta_1$ | $\delta_2$ | $\eta_1$ | $\eta_2$ | $\xi_1$ | $\xi_2$ | $\omega_1$ | $\omega_2$ |
|---|---|---|---|---|---|---|---|---|---|---|---|---|
| $a_1^B$ | ↗ | — | — | — | ↘ | ↘ | — | — | — | — | ↗ | ↗ |
| $a_2^B$ | — | ↗ | — | — | — | ↘ | — | — | — | — | — | ↗ |
| $s_{x1}^B$ | — | — | ↗ | — | ↘ | ↘ | ↗ | — | — | — | ↗ | ↗ |
| $s_{x2}^B$ | — | — | — | ↗ | — | ↗ | — | ↗ | — | — | — | ↗ |

Note:
↗ indicates a positive correlation with exogenous parameters, ↘ indicates a negative correlation with exogenous parameters, and the horizontal line indicates no correlation with exogenous parameters.

**Table 4 Sensitivity analysis of exogenous parameters in the vertical cooperative alliance model.**

| | $\theta_1$ | $\theta_2$ | $\alpha_1$ | $\alpha_2$ | $\delta_1$ | $\delta_2$ | $\eta_1$ | $\eta_2$ | $\xi_1$ | $\xi_2$ | $\omega_1$ | $\omega_2$ |
|---|---|---|---|---|---|---|---|---|---|---|---|---|
| $a_1^H$ | ↗ | — | — | — | ↘ | ↘ | — | — | — | — | ↗ | ↗ |
| $a_2^H$ | — | ↗ | — | — | — | ↘ | — | — | — | — | — | ↗ |
| $s_{M1}^H$ | — | — | ↗ | — | ↘ | ↘ | ↗ | — | ↘ | — | ↗ | ↗ |
| $s_{M2}^H$ | — | — | — | ↗ | — | ↘ | — | ↗ | — | ↘ | — | ↗ |
| $s_{N1}^H$ | — | — | ↗ | — | ↘ | ↘ | ↗ | — | — | — | ↗ | ↗ |
| $s_{N1}^H$ | — | — | — | ↗ | — | ↘ | — | ↗ | — | — | — | ↗ |

Note:
↗ indicates a positive correlation with exogenous parameters, ↘ indicates a negative correlation with exogenous parameters, and the horizontal line indicates no correlation with exogenous parameters.

Proposition 6 demonstrates that in the non-cooperative model, the platform's blockchain technology level is solely positively correlated with its own coefficient of influence on demand, both before and after innovation, and bears no correlation with the coefficient of influence of competitors' blockchain technology levels on their own demand.

**Proposition 7:** The sensitivity analysis of the level of blockchain technology regarding the key parameters in the vertical cooperative alliance model are $\frac{\partial u_{x1}^H}{\partial \lambda_1} > 0$, $\frac{\partial u_{M1}^H}{\partial \gamma_1} > 0$, $\frac{\partial u_{N1}^H}{\partial \gamma_1} = 0$, $\frac{\partial u_{x2}^B}{\partial \gamma_2} > 0$, $\frac{\partial u_{M2}^B}{\partial \gamma_2} > 0$, $\frac{\partial u_{N2}^B}{\partial \gamma_2} = 0$, respectively

Proposition 7 illustrates that at this juncture, the blockchain technology level of Platform M is positively correlated with its own influence coefficient on demand, as well as with the coefficient of the other party's blockchain technology level on its own demand.

Conversely, Platform N's blockchain technology level is exclusively positively correlated with its own influence coefficient on demand, independent of other factors. This reflects the fact that Platform N solely prioritizes its own profit maximization in decision-making and disregards a sense of cooperation.

## COMPARATIVE ANALYSIS

This section aims to examine the effects of product innovation and the three cooperative alliance models on each strategy. It builds upon the theoretical foundation established in the previous section and evaluates the correlation between the scale of each strategy and its profitability in different models.

**Proposition 8:** The size relationships between the innovative investment strategies of pre-innovation and post-innovation green suppliers, and the level of blockchain technology of the platforms under the three different decision-making models are $a_1^H > a_1^C = a_1^B$, $a_2^H > a_2^C = a_2^B$, $u_{M1}^H > u_{M1}^C > u_{M1}^B$, $u_{M2}^H > u_{M2}^C > u_{M2}^B$, $u_{N1}^C > u_{N1}^B = u_{N1}^H$, $u_{N2}^C > u_{N2}^B = u_{N2}^H$ respectively.

**Proposition 9:** Under the three different decision-making models, when $\alpha_1 \omega_2 \chi (1 + \phi) + \alpha_1 \omega_1 (r + \delta_2 - \xi_1 (r + \chi + \delta_1)(r + \delta_2) > 0$, $\alpha_2 \omega_2 - \xi_2 (r + \delta_2) > 0$ are satisfied, the size relationships of the innovation investment strategies of the pre-innovation and post-innovation platforms are $s_{M1}^H > s_{M1}^C > s_{M1}^B$, $s_{M2}^H > s_{M2}^C > s_{M2}^B$, $s_{N1}^C > s_{N1}^B = s_{N1}^H$, and $s_{N2}^C > s_{N2}^B = s_{N2}^H$.

**Proposition 10:** The relationship between the size of the green goodwill steady state before and after the innovation in the three different decision models are $G_{1\infty}^H > G_{1\infty}^C > G_{1\infty}^B$, $G_{2\infty}^H > G_{2\infty}^C > G_{2\infty}^B$.

**Proposition 11:** The total profit of green supply chain under horizontal cooperative alliance, non-cooperative and vertical cooperative alliance modes are denoted by $V^C$, $V^B$ and $V^H$ respectively. The relationship between the size of the total profit of green supply chain under different decision-making modes is $V^H > V^C > V^B$.

Propositions 8–11 suggest that horizontal and vertical cooperative alliance strategies can generally improve green goodwill, innovation investment strategy, blockchain technology level, and total green supply chain profit. The vertical cooperative alliance, however, offers a more significant improvement effect than its horizontal counterpart. Both models contribute to enhancing blockchain technology levels, but the vertical cooperative model has a stronger impact.

## RESULTS: NUMERICAL EXAMPLE

We will now enhance the conclusions from the previous section by providing numerical examples and examining the effects of innovation realization rate and enhancement rate on green goodwill and system profit. Additionally, we will analyze how different cooperative alliance modes can improve profit, thereby yielding valuable managerial insights. In order to achieve this, we have established system parameters based on *Guo et al. (2021)* and *Shen, Dong & Minner (2022)* and tailored them to the specific context of this article.

$r = 0.1, \theta_1 = 0.7, \theta_2 = 1, \alpha_1 = 0.8, \alpha_2 = 1, \delta_1 = 0.5, \delta_1 = 0.3, \omega_1 = 0.7, \omega_2 = 1, \eta_1 = 0.7,$
$\eta_2 = 1, \xi_1 = 0.5, \xi_1 = 0.6, \lambda_1 = 0.8, \lambda_2 = 1, \rho_{F1} = 2, \rho_{F2} = 3, \rho_{M1} = 2, \rho_{M2} = 3, \rho_{N1} = 2,$
$\rho_{N2} = 3, k_F = 1.2, K_{Ms} = 1.5, K_{Ns} = 1.5, K_{Mu} = 1.5, K_{Nu} = 1.5, \chi = 0.6, \phi = 0.6$

## Plot of the time trajectory of green goodwill in different scenarios

Figure 4 provides insight into several key points. Firstly, managers must differentiate between scenarios with dissimilar realization rates and varying enhancement rates when predicting innovation outcomes. This is due to the presence of unpredictable factors or varying levels of risk inherent in product development and innovation. Secondly, it is clear that green goodwill experiences an escalation and stabilization over time, both before and after the innovation phase. Thirdly, successful product innovation leads to a significantly higher steady-state value of green goodwill compared to the pre-innovation equilibrium. This is exemplified by the recent announcement by Corona, the global brand under AB InBev, on May 11, 2022, as reported by PRNewswire, regarding the launch of Corona Tropical.

## Comparison of the total profit of the two platforms under different scenarios

Figure 5 shows that successful innovations can positively impact platforms, resulting in increased profits and improved profitability through horizontal alliances. Collaborative partnerships can be particularly effective in achieving Pareto improvement, especially when blockchain technology generates positive spillover effects.

## Comparison of total profit of green supplier BB and platform MM under different scenarios

In this section, we examine the consolidation of profits between the green supplier and one of the platforms to evaluate the effects of different alliance strategies on corporate profits. As shown in Fig. 6: (1) The joint profits of both the green supplier and the platform increase and level off over time, with successful innovation resulting in a significant rise in benefits. (2) Vertical cooperative alliances have the potential to achieve Pareto improvements in the profits of both parties relative to non-cooperative models.

## Comparison of total profitability of green supply chains under different scenarios

Based on Fig. 7, we can see that: (1) Over time, the total profit of the green supply chain consistently improves and eventually stabilizes, regardless of the rates of product innovation realization and promotion. Successful product innovation leads to an overall increase in profit. (2) When the realization rate and promotion rate are both high, the system's total profit is significantly boosted. (3) Regardless of whether innovation has taken place or not, both the horizontal and vertical cooperative alliance models lead to a higher total profit for the green supply chain system compared to the non-cooperative mode. This improvement in profit benefits all parties involved. The vertical cooperative

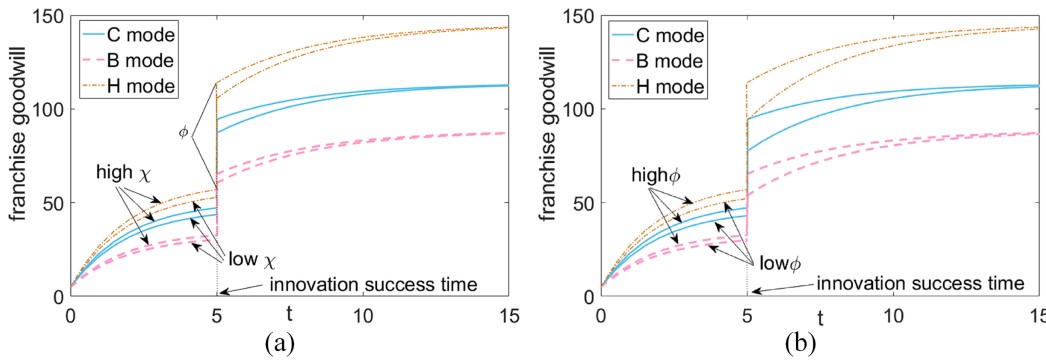

**Figure 4 Green goodwill in different situations.** (A) Lift rate fixation—green goodwill at different realization rates. (B) Fixed realization rate—platform reputation at different lift rates.

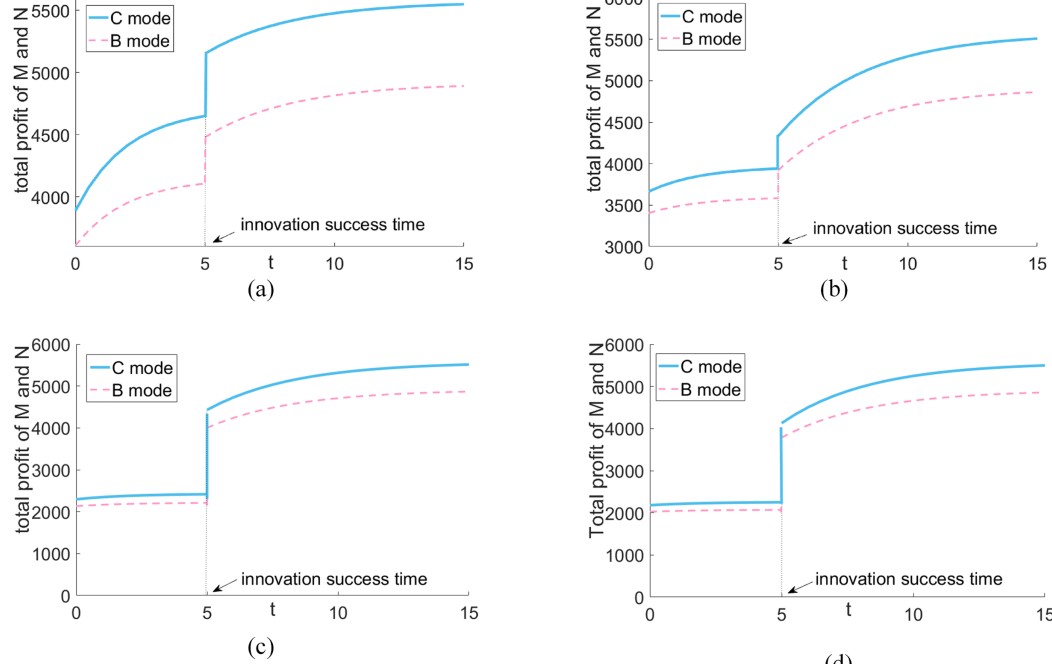

**Figure 5 Comparison of the total profit of the two platforms under different circumstances.** (A) High realization rate—high promotion rate. (B) High realization rate—low uplift. (C) Low realization rate—high uplift rate. (D) Low realization rate—low promotion rate.

alliance model has a stronger impact on the system's total profit compared to the horizontal cooperative alliance model.

## Time sensitive validation

With the rapid advancements in science and technology, coupled with the growth of the Internet economy, computer information technology has become ubiquitous in various industries, particularly in blockchain and artificial intelligence. As society progresses and

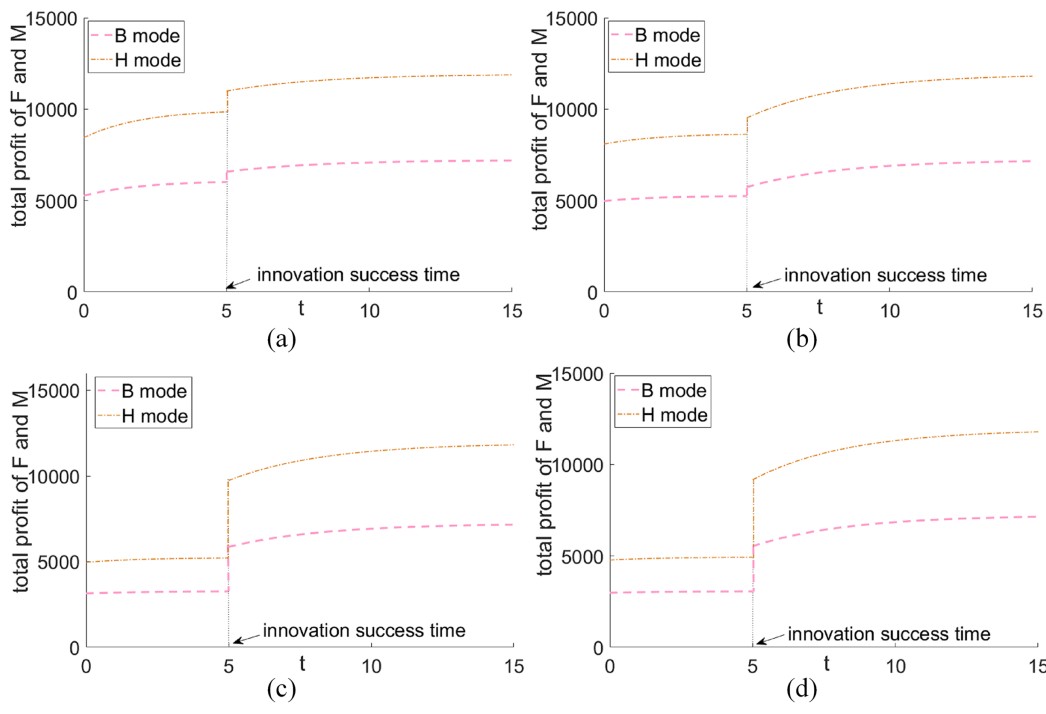

**Figure 6 Comparison of total profit between green supplier F and platform M under different circumstances.** (A) High realization rate—high promotion rate. (B) High realization rate—low uplift. (C) Low realization rate—high uplift rate. Low realization rate—low promotion rate.

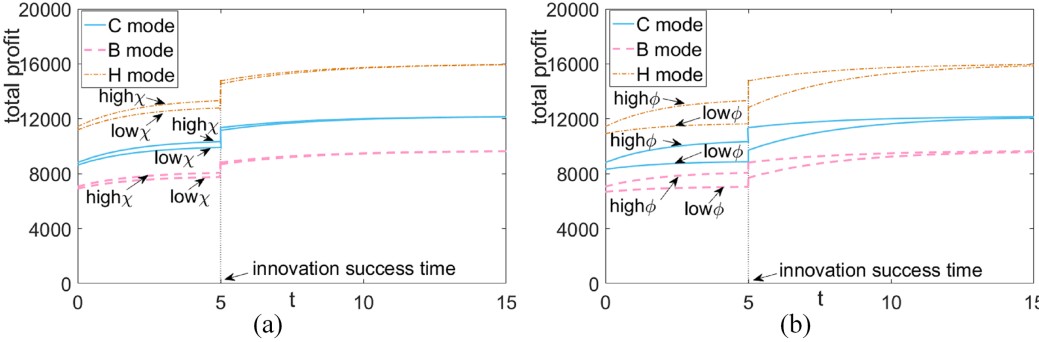

**Figure 7 Total profit comparison of green supply chain in different situations.** (A) Lift rate fixation—green goodwill at different realization ratesv. (B) Realization rate fixed-platform reputation at different lift rates.

times change, the traditional mode of information transmission has become inadequate to meet societal needs. This has, in turn, fostered the development of Internet information technology and further expanded the reach of computer information technology. To assess the potential impact of emerging technology on supply chain profits, this subsection examines the time sensitivity of total profits in a green supply chain, as shown in Table 5.

According to the data presented in Table 5, the overall profit of the green supply chain is closely tied to the passage of time. As time passes, the profit steadily rises until it reaches a

**Table 5 Sensitivity validation of total profit in green supply chain.**

| Time | Mode | | | | | | |
|---|---|---|---|---|---|---|---|
| | 0 | 5 | 10 | 20 | 40 | 200 | 600 |
| C | 8.2815e+03 | 9.1241e+03 | 9.1933e+03 | 9.1994e+03 | 9.1995e+03 | 9.1995e+03 | 9.1995e+03 |
| B | 6.6429e+03 | 7.1917e+03 | 7.2368e+03 | 7.2408e+03 | 7.2408e+03 | 7.2408e+03 | 7.2408e+03 |
| H | 1.0781e+04 | 1.1845e+04 | 1.1932e+04 | 1.1940e+04 | 1.1940e+04 | 1.1940e+04 | 1.1940e+04 |

**Table 6 Data comparison.**

| Code | Short name | Statistical deadline | Net profit | R&D investment | Innovation investment strategy |
|---|---|---|---|---|---|
| 000050 | Deep Tenma A | 2020-12-31 | 1,474,521,450.1 | 2,065,755,595.7 | 2,211,782,175.1 |
| 000050 | Deep Tenma A | 2021-12-31 | 1,542,457,101.1 | 2,068,112,985.8 | 2,313,685,651.6 |
| 000050 | Deep Tenma A | 2022-12-31 | 107,794,661.5 | 2,983,239,544.0 | 161,691,992.2 |
| 000066 | Great Wall of China | 2020-12-31 | 974,585,489.9 | 1,116,471,762.2 | 1,461,878,234.8 |
| 000066 | Great Wall of China | 2021-12-31 | 682,488,877.3 | 1,383,301,877.5 | 1,023,733,315.9 |
| 000066 | Great Wall of China | 2022-12-31 | 226,275,882.0 | 1,501,305,559.5 | 339,413,823.0 |
| 000801 | Jiuzhou in Sichuan province | 2021-12-31 | 180,772,229.6 | 298,405,476.8 | 271,158,344.4 |
| 000801 | Jiuzhou in Sichuan province | 2022-12-31 | 241,425,432.6 | 325,414,111.6 | 362,138,148.9 |
| 000810 | Skyworth Digital (Hong Kong company) | 2020-12-31 | 354,102,471.7 | 517,101,353.3 | 531,153,707.5 |
| 000810 | Skyworth Digital (Hong Kong company) | 2022-12-31 | 805,849,761.8 | 622,845,594.2 | 1,208,774,642.7 |

stable level. Once the steady state is achieved, time no longer has an impact on the profit of the green supply chain. These findings align with those of the previous section and support the conclusion that decision makers should prioritize pre-innovation profits.

## CONCLUSIONS

This article delves into the impact of uncertainty in innovation realization on the performance and optimal operation strategies of green supply chain members in a blockchain-enabled setup. We also consider the Pareto improvement effect of various cooperative alliance modes on overall system profit. Using differential game theory, we solve for the optimal strategies and system profits of green supply chain members under different decision-making modes. Additionally, we reveal the influence of key exogenous parameters on equilibrium strategies through sensitivity analysis. We conduct comparative analysis to examine the optimal strategies, green goodwill, and profit steady state sizes of different decision-making modes (horizontal cooperative alliance mode, non-collaborative mode, and vertical cooperative alliance mode), as well as the impact of innovation realization and enhancement rates on the total profit of the system. Finally, we verify the effects of innovation realization and enhancement rates on platform reputation, system profit, and the Pareto improvement effect of different decision-making modes on total system profit through numerical examples. The study's main conclusions highlight the

importance of considering uncertainty in innovation realization and the benefits of cooperative alliance modes in maximizing overall system profit.

(1) When decision makers have an idea of the probability of innovation coming to fruition, they adjust their priorities and become more motivated to pursue pre-innovation profits over post-innovation profits. Additionally, predictions of innovation events impact pre-innovation strategies, leading green suppliers to invest more in product research and development, and platforms to promote higher levels of green goodwill through innovation investments. This creates a beneficial cycle of positive outcomes as innovation likelihood and upliftment increase.

(2) The success of innovation and its impact on environmental goodwill and profit margin are interrelated. The higher the realization rate and enhancement rate, the greater the pre-innovation green goodwill, which in turn affects the post-innovation goodwill. Additionally, the size of the enhancement rate plays a direct role in the growth of green goodwill post-innovation. The impact on profit margin follows a similar pattern, with the level of realization rate and enhancement rate determining the profitability of the green supply chain both before and after innovation.

(3) Regardless of the timing of innovation, both the horizontal and vertical cooperative alliance models can enhance the overall profitability of the green supply chain compared to the non-cooperative model. This leads to an effective Pareto improvement in the total system profit. The Pareto improvement effect of the two cooperative alliance models on the system's profit increases with the possibility of realizing the innovation and the enhancement rate. The vertical cooperative alliance model has a stronger improvement effect on the total system profit compared to the horizontal cooperative alliance model, especially at a high realization rate and enhancement rate.

(4) When green suppliers opt for a vertical cooperative alliance with a platform, their degree of innovative investment strategy tends to increase. Meanwhile, the alliance between platforms does not impact their investment strategy. For platforms, both horizontal and vertical cooperative alliances incentivize them to enhance their blockchain technology level. However, the promotion effect of vertical alliances surpasses that of horizontal alliances. Furthermore, the operating mode chosen by platforms before and after innovation greatly impacts their investment strategies. Vertical alliances between platforms and green suppliers are more effective in promoting innovation investment strategies.

## APPLICABILITY AND LIMITATIONS OF THE FINDINGS

This research article delves into the effects of product innovation on decision-making within a green supply chain. It distinguishes between pre- and post-innovation environments, analyzing how strategy formulation options and firm profits differ between the two. Additionally, the article explores the impact of various cooperative alliance models on profits within the green supply chain and how the success of innovation influences this impact. To provide a more accurate representation of the problem at hand and to align

with real-world scenarios, the article employs a differential game model in conjunction with the randomized stopping problem. It is worth noting that the model presented in this article can be applied to other industry sectors, including e-commerce supply chain management, where e-commerce platforms operate under two sales modes: resale and agency sales. By studying the impact of product innovation on the strategies of e-commerce supply chain members, we can expand the applicability of the model to different industries.

It is important to note that this study's model has some limitations, and the article solely focuses on a one-way green supply chain, which has its own set of restrictions. With the growing emphasis on environmental protection, the government, corporations, and consumers alike have become increasingly invested in the green movement. However, many discarded products and packaging materials ultimately end up as domestic waste, and most waste disposal methods still rely on traditional landfills that occupy vast amounts of land. This leads to pest infestations, sewage overflow, and a pervasive odor that severely pollutes the environment. Closed-loop recycling offers a solution that can reduce waste treatment and equipment costs, minimize land resource consumption, and yield social, economic, and ecological benefits. As such, future research should consider exploring the implementation of blockchain-enabled closed-loop supply chains and developing strategies for recycling green products.

## EMPIRICAL TESTING—CASE STUDIES

To support the theoretical model with concrete evidence and ensure its practicality, we analyzed data from 2,698 enterprises between 2020–2022. We examined factors such as net profit, R&D investment, expenses, development costs, and overall profit to conduct empirical research. Our goal was to determine if the model could be applied over an extended period. We used the model's strategy to develop an innovation investment plan for each company post-innovation and compared it to actual R&D investment amounts. For brevity, we present a comparison of data from 10 enterprises in Table 6.

According to Table 6, our empirical test of real case data reveals that the innovation investment levels predicted by our study align with those of most enterprises in the dataset, confirming the validity of our article's model. However, there are exceptions, such as "Shenzhen Tianma A" whose actual R&D investment in 2022 was 2,983,239,544.0, while our model predicted an investment of 161,691,992.2. This indicates that the enterprise may have wasted resources, as investing only 161,691,992.2 would have resulted in a net profit of 107,794,661.5 for the year. Further calculations show that if the enterprise had invested 2,983,239,544.0, the expected net profit would have been 1,988,826,362.7, whereas the actual net profit for the year was only 107,794,661.5. The same situation occurred with the Great Wall of China's R&D investment strategy in 2022. We hope that our model can provide theoretical support and advice for companies facing similar challenges and decision-making problems.

### Funding

This work was funded by the China University Innovation Fund (2021LDA11003), the Capital University of Economics and Business 2023 Science and Technology Innovation Project: Optimization Research of Green Supply Chain Innovation and Investment Strategies under Blockchain Empowerment, and the 2024 School-level Teaching Reform Project of Capital University of Economics and Business: Artificial Intelligence Talent Cultivation and Teaching Practice Integrating "Artificial Intelligence + X". The funders had no role in study design, data collection and analysis, decision to publish, or preparation of the manuscript.

### Grant Disclosures

The following grant information was disclosed by the authors:
China University Innovation Fund: 2021LDA11003.
Capital University of Economics and Business 2023 Science and Technology Innovation Project: Optimization Research of Green Supply Chain Innovation and Investment Strategies under Blockchain Empowerment.
2024 School-level Teaching Reform Project of Capital University of Economics and Business: Artificial Intelligence Talent Cultivation and Teaching Practice Integrating "Artificial Intelligence + X".

### Competing Interests

The authors declare that they have no competing interests.

### Author Contributions

- Fangfang Guo conceived and designed the experiments, performed the experiments, analyzed the data, performed the computation work, prepared figures and/or tables, authored or reviewed drafts of the article, and approved the final draft.
- Zhuang Wu performed the experiments, analyzed the data, prepared figures and/or tables, and approved the final draft.
- Yuanyuan Wang performed the computation work, prepared figures and/or tables, authored or reviewed drafts of the article, and approved the final draft.
- Chenjun Liu performed the computation work, prepared figures and/or tables, authored or reviewed drafts of the article, and approved the final draft.

### Data Availability

The raw data and procedure for proving all the propositions are available in the Supplemental File.

### Supplemental Information

Supplemental information for this article can be found online at http://dx.doi.org/10.7717/peerj-cs.2002#supplemental-information.

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
