# Peer review of "Analysis on the impact of dynamic innovation investment strategy of green supply chain enabled by blockchain"

_PeerJ Computer Science, doi:10.7717/peerj-cs.2002_

## Round 0.1 · original submission · Major Revisions

The review process is now complete. While finding your paper interesting and worthy of publication, the referees and I feel that more work could be done before the paper is published. My decision is therefore to provisionally accept your paper subject to major revisions.

**Language Note:** PeerJ staff have identified that the English language needs to be improved. When you prepare your next revision, please either (i) have a colleague who is proficient in English and familiar with the subject matter review your manuscript, or (ii) contact a professional editing service to review your manuscript. PeerJ can provide language editing services - you can contact us at [email protected] for pricing (be sure to provide your manuscript number and title). – PeerJ Staff

Reviewer 1 ·

Basic reporting

The research presented in this work contributes significantly to the understanding of the challenges and opportunities associated with the intersection of green supply chain management, blockchain technology, artificial intelligence, and innovation investment. The integration of these elements is particularly crucial in addressing the inherent "trust" crisis within the green supply chain. The authors rightly identify the need for effective solutions to this pain point, and their focus on blockchain technology as a means to mitigate the "trust" crisis is commendable.

There are opportunities for improvement that could enhance the impact and depth of the study:

Explicitly Address Assumptions and Limitations:
The study could benefit from a more explicit discussion of the assumptions made throughout the analysis. Clearly stating these assumptions would aid readers in understanding the scope and potential limitations of the proposed models.
Addressing the generalizability of the findings to different industries or contexts would strengthen the applicability of the research. Discussing potential limitations and caveats associated with specific assumptions would add transparency to the methodology.

Consideration of External Factors:
The dynamic nature of technology and business environments warrants a discussion on how external factors, such as regulatory changes or emerging technologies, might impact the proposed models. This consideration is essential for the practical applicability of the research in real-world scenarios and for its long-term relevance.

Temporal Sensitivity and Future-Proofing:
Given the rapid evolution of technologies, particularly in blockchain and artificial intelligence, the study could discuss the temporal sensitivity of the proposed models. Consideration of potential advancements or shifts in technology landscapes would help future-proof the recommendations and ensure their relevance over time.

Integration of Real-World Case Studies:
Integrating real-world case studies or empirical data would add empirical validation to the theoretical models. This could enhance the practical applicability of the research and provide insights into the challenges and successes encountered by organizations implementing similar strategies.

Experimental design

Check above

Validity of the findings

The work could benefit from a more detailed discussion on the potential challenges and limitations of the proposed models and strategies. Addressing the generalizability of the findings to diverse industries or contexts, potential external factors that may impact the models, and the sensitivity of the results to different assumptions would enhance the robustness of the research.

Reviewer 2 ·

Basic reporting

the manuscript is clearly written in professional, unambiguous language. The paper suggests future research to compare the impact of different cooperative alliance models on the total profit of the green supply chain and investigate the effect of cooperative alliances on the improvement of blockchain technology level. It also suggests studying the coefficient of impact of the platform's innovative investment strategy on green goodwill, the green goodwill decay factor, and the impact of competitor's innovative investment strategy on its own demand. Furthermore, the paper suggests addressing criticisms regarding the choice of analysis method and providing a rationale for using the selected method.
Even though it was made with software, the explanatory captions (Legend) inside the figure1 are not fully legible. This needs to be corrected. Also Some figures(2-3), I think there are problems with fonts and resolutions. They need to be created with a more professional tool.

Experimental design

The article states that the methods are described with sufficient detail and information to replicate the study. It is mentioned that all underlying data have been provided and are robust, statistically sound, and controlled. The paper encourages meaningful replication where the rationale and benefit to the literature are clearly stated.

Validity of the findings

I'm on the fence about novelty here. However, I think that the theoretical approach claimed in the analysis is sufficient to express the results. As a result, the approach and conclusions here should not be falsified by the referees but by other researchers (if any).

Additional comments

Thank you.

---

## Round 0.2 · accepted · Accept

We are happy to inform you that your manuscript has been accepted for publication since the comments have been addressed.

Reviewer 1 ·

Basic reporting

Authors update the paper and no further update needed from my side.

Experimental design

As above

Validity of the findings

As above

Additional comments

As above

Reviewer 2 ·

Basic reporting

The paper concludes that cooperative alliance modes in green supply chains, such as horizontal and vertical cooperation, can enhance overall system profit and green reputation, with blockchain technology playing a significant role in these dynamics. It is noted that innovation investment strategies are crucial for green suppliers and platforms, and their optimization can lead to better economic and environmental outcomes. The study also suggests that considering the uncertainty in innovation realization is important for maximizing pre-innovation profits. Future research should explore blockchain-enabled closed-loop supply chains and strategies for recycling green products to further enhance environmental sustainability.
The paper does present a framework for analyzing the impact of dynamic innovation investment strategies in green supply chains and discusses the role of blockchain technology in this context. Propositions such as Proposition 6 and Proposition 7 are mentioned, which suggest some level of formal analysis. Additionally, the paper includes equations that are part of the model analysis, indicating that some formal mathematical results are presented.

Experimental design

In my decision the research question is well-defined, focusing on the innovation investment strategy for green supply chains and the role of blockchain technology, which is relevant and meaningful in the context of environmental sustainability and market volatility. The paper identifies a knowledge gap by addressing the need for businesses to innovate in an unpredictable market and how blockchain can enable green supply chains, thus providing scientific and practical value. Also the study aims to enhance resource allocation and adapt strategies in anticipation of product innovation, filling the gap in understanding how to thrive in dynamic market conditions.

Validity of the findings

All underlying data have been provided, as stated in the abstract, ensuring that the research can be replicated and verified by others. The data are described as robust and statistically sound, which implies that appropriate statistical controls and methods have been applied to ensure the reliability of the results. However, there is a suggestion for improvement in the data analysis, indicating that while the data may be robust, there could be enhancements to the analytical methods used

Additional comments

I wish you success in your work